# Remote Sensing for Property Valuation: A Data Source Comparison in Support of Fair Land Taxation in Rwanda

**Mila Koeva** [1,*], **Oscar Gasuku** [2], **Monica Lengoiboni** [1], **Kwabena Asiama** [3], **Rohan Mark Bennett** [4,5], **Jossam Potel** [6] and **Jaap Zevenbergen** [1]

1 Faculty of Geo-Information Science and Earth Observation (ITC), University of Twente, Drienerlolaan 5, 7522 NB Enschede, The Netherlands; m.n.lengoiboni@utwente.nl (M.L.); j.a.zevenbergen@utwente.nl (J.Z.)
2 Director of Infrastructure, Osc Infrastructures Private Limited, Rubavu District, Gisenyi 173, Rwanda; oscdirector@rubavu.gov.rw
3 Geodetic Institute, Leibniz Universität Hannover, Nienburger Strasse 1, 30167 Hannover, Germany; asiama@gih.uni-hannover.de
4 Department of Business Technology and Entrepreneurship, School of Business, Law and Entepreneurship, Swinburne University of Technology, John St, Hawthorn, VIC 3122, Australia; rohanbennett@swin.edu.au
5 Kadaster Netherlands, The Dutch Cadastre, Land Registry, and National Mapping Agency, 7311 KZ Apeldoorn, The Netherlands
6 Department of Land Administration and Management, Faculty of Applied Fundamental Sciences, INES, Musanze, NM Street, Ruhengeri 155, Rwanda; jossam2003@ines.ac.rw
* Correspondence: m.n.koeva@utwente.nl; Tel.: +31-534-874-410

**Abstract:** Remotely sensed data is increasingly applied across many domains, including fit-for-purpose land administration (FFPLA), where the focus is on fast, affordable, and accurate property information collection. Property valuation, as one of the main functions of land administration systems, is influenced by locational, physical, legal, and economic factors. Despite the importance of property valuation to economic development, there are often no standardized rules or strict data requirements for property valuation for taxation in developing contexts, such as Rwanda. This study aims at assessing different remote sensing data in support of developing a new approach for property valuation for taxation in Rwanda; one that aligns with the FFPLA philosophy. Three different remote sensing technologies, (i) aerial images acquired with a digital camera, (ii) WorldView2 satellite images, and (iii) unmanned aerial vehicle (UAV) images obtained with a DJI Phantom 2 Vision Plus quadcopter, are compared and analyzed in terms of their fitness to fulfil the requirements for valuation for taxation purposes. Quantitative and qualitative methods are applied for the comparative analysis. Prior to the field visit, the fundamental concepts of property valuation for taxation and remote sensing were reviewed. In the field, reference data using high precision GNSS (Leica) was collected and used for quantitative assessment. Primary data was further collected via semi-structured interviews and focus group discussions. The results show that UAVs have the highest potential for collecting data to support property valuation for taxation. The main reasons are the prime need for accurate-enough and up-to-date information. The comparison of the different remote sensing techniques and the provided new approach can support land valuers and professionals in the field in bottom-up activities following the FFPLA principles and maintaining the temporal quality of data needed for fair taxation.

**Keywords:** property valuation; property taxation; remote sensing; land; UAV

## 1. Introduction

Remote sensing (RS) data has been shown to be efficient in obtaining precise spatial information for a variety of applications, which is crucial to achieving sustainable development goals (SDGs). Property ownership, value, and rights are included as sub-goal 1.4 of the SDGs. Nowadays, being part of the fourth industrial revolution, remotely sensed images are ubiquitous in many socio-economic endeavors. Therefore, the use of remotely sensed data is highly advised for use in fit-for-purpose land administration (hereafter

FFPLA) [1]. Moreover, contemporary developments in photogrammetry and computer vision, coupled with high-resolution remote sensing data, has led many researchers to explore the use of machine learning to extract information automatically from images for cadastral applications [2].

Satellite images in various spatial and temporal resolutions are globally available, making them very useful for monitoring daily dynamics. Dabrowski and Latos [3] investigated the applicability of remote sensing images for land-related applications focusing on the effect of the different spatial, radiometric, temporal, and spectral resolution. Haeusler, Gomez, and Enßle [4] and Ali and Deininger [5] showed that remote sensing data, especially high-resolution satellite imagery (HRSI), can be used to extract or measure the height of buildings, which is useful for urban planning, assessment of property taxes, estimation of floor area, and so on. Jain [6] acquired socioeconomic attributes, roof material, shape, structure of buildings, and the age of construction from high-resolution imagery using object-based classification for the purpose of property taxation.

However, for deriving precise property characteristics needed for valuation for taxation purposes, higher spatial resolution is preferable. Recently, unmanned aerial vehicles (UAVs) have proven to be promising for many applications such as agriculture [7], mapping [8], surveying and cadaster applications [9–12], architecture and archeology [13], cultural heritage [14], among others. However, to the best of the authors' knowledge, the applicability of UAV images for property valuation has not been examined empirically [15]. Therefore, the current study aims to assess and compare different remote sensing data for property valuation for taxation, focusing on Rwanda as a case study.

Property valuation is done for different purposes, including taxation, sales, and insurance, amongst others. Valuation is a process of estimating the amount for which a property will be exchanged or the amount of taxes that should be paid for it on a particular date [16,17]. Being an art and a science, the complexity of the valuation process covers, among others, transparency of the market, diverse purposes, and stakeholders involved [16,18]. It is further strongly influenced by the background and experience of the valuer, as well as the global trend of economic development and investment interests [19,20].

There are two types of valuation procedures. As stated by Wallace and Williamson [21], the first is the valuation of a single property, and the second is so-called mass valuation, which refers to an area combining many properties. Valuations for taxation are usually carried out through a mass valuation. Property taxes are defined as the amount of money levied on a person, natural or unnatural, by a government, for the holding of real estate within a particular jurisdiction. Different approaches have been developed worldwide for the valuation of property for taxation, including artificial neural networks (ANNs), hedonic pricing methods, spatial analysis methods, and others [22].

Property tax collection approaches differ from country to country. However, the two main approaches are area-based taxation and value-based [23]. Typically, area-based taxations are used for determining the assessed value of the property where the property markets are not mature enough to support a value-based system [24]. For area-based tax, the taxes build on the unit, and therefore the unit assessment must be accounted for in the rate [25]. The rate is levied per $m^2$ of land area, building area, or a combination of the two. The unit's value reflects several factors such as the property's location, accessibility, land use/zoning, laws and regulations, socioeconomics, condition, age, and neighborhood development. Value-based taxation is based on market value (capital or rental).

There are many factors that affect property value such as locational, physical, legal, and economic factors [18,26]. They can be split into two groups, internal and external factors (Figure 1). This research is focused on the use of remotely sensed data for extraction of physical attributes or geospatial factors (e.g., parcel area, built-up area/gross floor area) and locational factors (e.g., accessibility, neighborhood development, and environment).

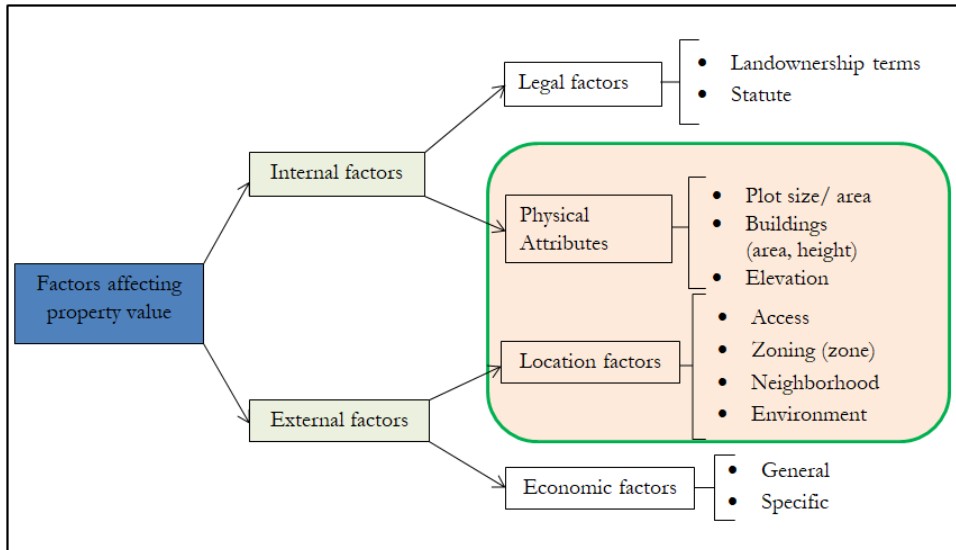

**Figure 1.** Factors affecting property value (Source: adapted based on [26]).

Rwanda has both property tax and a land lease fee, both of which vary depending on the tax base. The property tax is determined based on the open market value. In Rwanda, professional valuers prepare the valuation report of the property (parcel, buildings, and improvement on it). The tax amount is 0.1% of the total value of the property for industrial buildings, 1% for residential properties, and 0.5% for commercial buildings [27]. However, the land lease fee is based on the $m^2$ rate determined by the district [28,29]. The property tax relies on the percentage of the open market value of the property [30]. Both single property and mass valuation are used to assess the open market value of the property [21].

In 2008 and 2009, a traditional aerial survey with a 3000 m flying height was executed over the territory of Rwanda with a digital photogrammetric camera Vexcel UltraCamX. Post-processing procedures were completed by a Dutch photogrammetric company [31]. A digital elevation model and an orthophoto with a spatial resolution of 22 cm were further produced for the entire country [31]. The Rwandan land cadastre was built based on the orthophotos from these aerial and satellite images. Local citizens employed and mainly trained as 'para-surveyors' delineated the parcel limits on the imagery printouts that were scanned, geo-referenced, and digitized. Currently, property valuation relies on the data provided by the Rwanda land use management authority (RLMUA), especially the parcel area, the location of the property, and the land use; all this information can be obtained from an inspection of the land title and also from a field visit. For instance, the Rwanda National land-use masterplan, as well as the subsequent master plan for Kigali city, were developed based on the generated orthophotos from aerial images. These are currently sources of property valuation data [32]. However, substantial changes since 2009 have seen the database missing information such as buildings, improvements, accessibility, conditions, and zoning [33]. Therefore, methods for regular updating of the cadastral and valuation information are of high importance.

In summary, the usage of RS data for property valuation for taxation is still exploring innovations in the land administration domain, particularly as higher resolution data becomes cheaper and easier to capture more frequently. Therefore, the current research aims to assess and compare three different remote sensing technologies for property valuation for taxation, focusing on Rwanda as a case study, and ultimately to propose a new UAV-based approach. In alignment with the FFPLA principles, the study aims to glean lessons for that specific country context, but also for RS application in the domain more generally.

In the next section, the materials and methods used are presented, framed as a comparative analysis of different datasets over the case context of Rwanda. Results are presented

first with regard to the data requirements for property valuation in Rwanda, and then an assessment of the different datasets against those requirements. The discussion and conclusion focus on discerning the immediacy of application of the results in Rwanda, and particularly the identification of the most relevant data source for property valuation for taxation purposes in Rwanda. Attention is also given to broader applications beyond Rwanda and further research requirements.

## 2. Materials and Methods

### 2.1. Study Area

The study area of Nyarutarama cell-Remera sector Gasabo districts (Figure 2) was selected for the current research. This area witnessed many changes over the years, which are reflected in differences in the values of the properties.

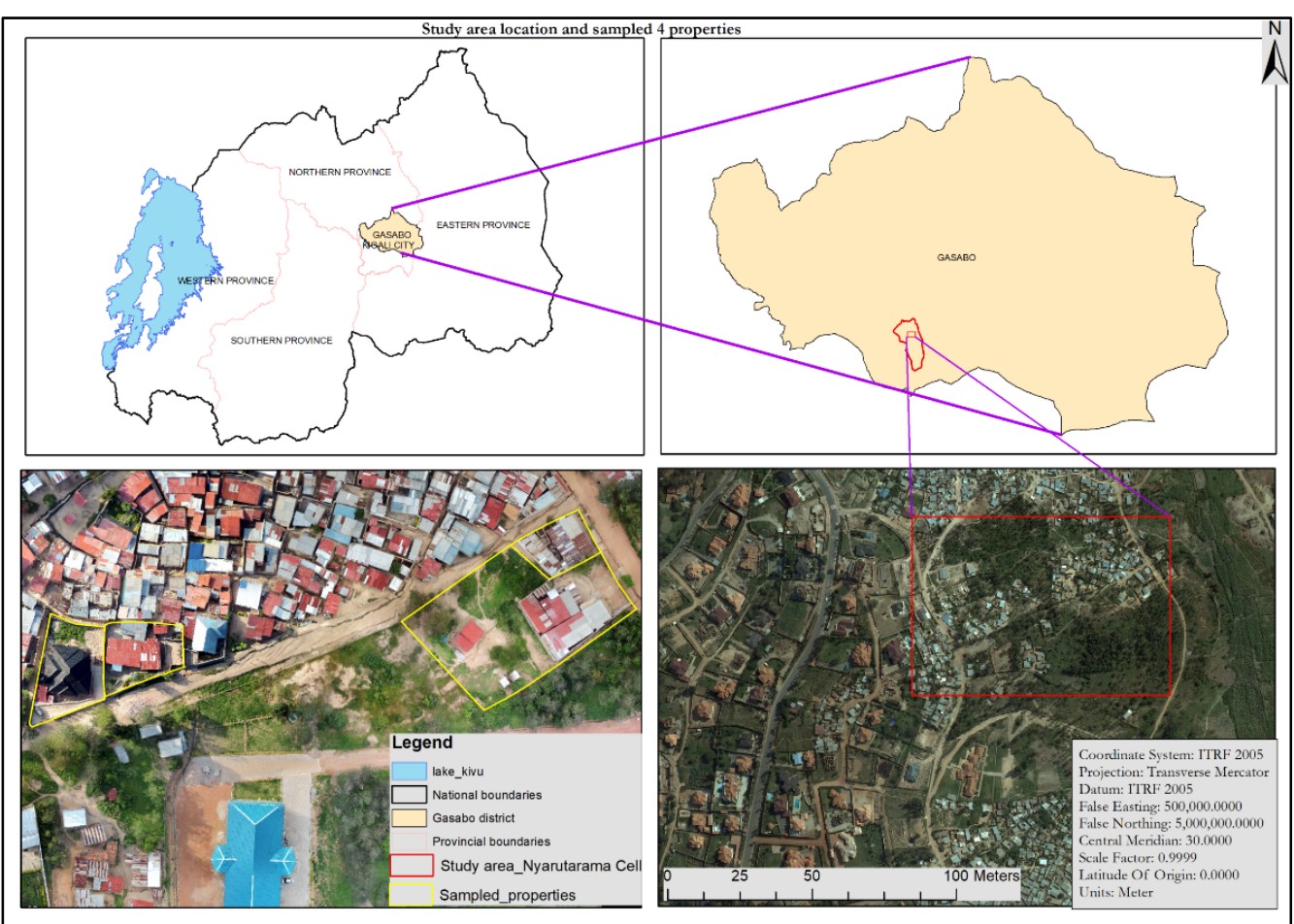

**Figure 2.** Location of the study area.

### 2.2. Data Sources

As shown in Table 1 and Figure 3 below, the orthophoto based on aerial images acquired in 2008/2009 with a digital camera on board an airplane was provided by the RLMU, and the satellite image was obtained from the Image repository of the University of Twente (ITC). The UAV images used for this research were collected in 2015 with a DJI Phantom 2 Vision Plus quadcopter at a flying height of 50 m. For the study, a total of 1172 geotagged nadir images were captured, and only 954 were obtained. However, an average of 85% forward and 75% side-overlaps was achieved. An orthophoto covering 950 m$^2$ with a spatial resolution of 3.3 cm and a radiometric resolution of 8 bits was produced with Pix4DCapture software. The final orthophoto with a positional accuracy of

6.0 cm was produced based on 13 premeasured ground control points with Leica GNSS with an accuracy of 2 cm [8].

**Table 1.** Used datasets and their sources.

| Dataset | Source | Acquisition Date | Spatial Resolution | Radiometric Resolution | Spectral Resolution |
|---------|--------|------------------|--------------------|------------------------|---------------------|
| Orthophoto from airplane aerial images | RLMU | 2008/2009 | 22 cm | 8-bit | 3 bands |
| Satellite Worldview2 image | ITC image repository | 2013 | 50 cm | 16-bit | 4 bands |
| Orthophoto from aerial UAV images | ITC image repository | 2015 | 3.3 cm | 8-bit | 4 bands |

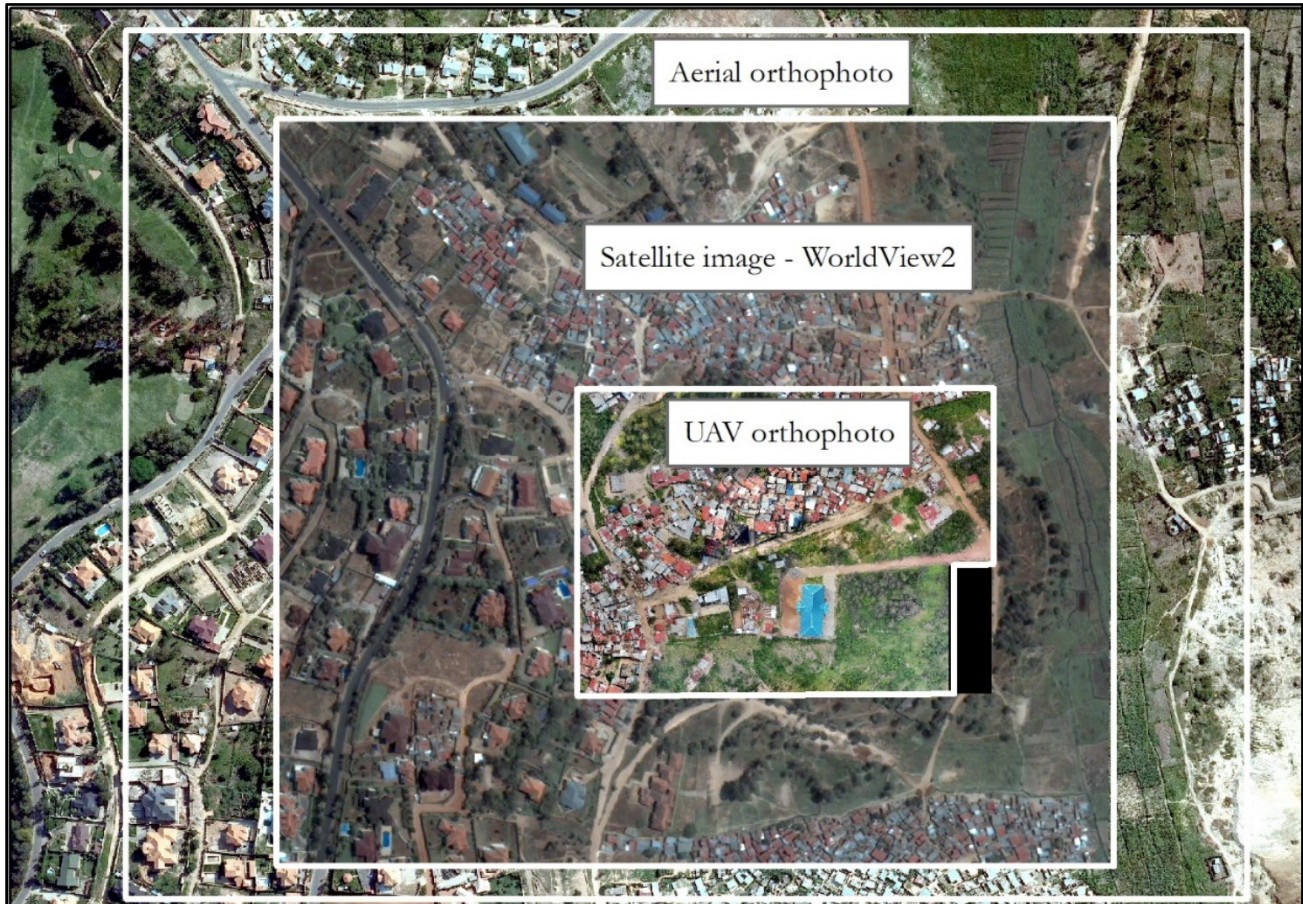

**Figure 3.** Coverage of the remote sensing data in Nyarutarama cell.

In this research, a mixed-methods approach combining quantitative and qualitative methods were used. The qualitative and quantitative data were given equal weighting and considered to be captured in parallel. Prior to the field visit, the concepts of property valuation for taxation and remote sensing were reviewed (Figure 4).

First, in terms of quantitative data collection, using high precision GNSS (Leica), reference data during the field survey was collected to be used for quantitative assessment of the extracted spatial features and property boundaries based on the images.

Second, in terms of qualitative data, primary data was collected via semi-structured interviews [34], focus group discussions, and field surveys aiming to assess the different RS data for valuation purposes. This interview technique was used to gather social perceptions and included the presentation of oral–verbal stimuli and replies [35]. In this research, face to face personal interviews were also completed. The experts sampling method [36] was used for the selection of the respondents based on their organization and functions, as shown in Table 2.

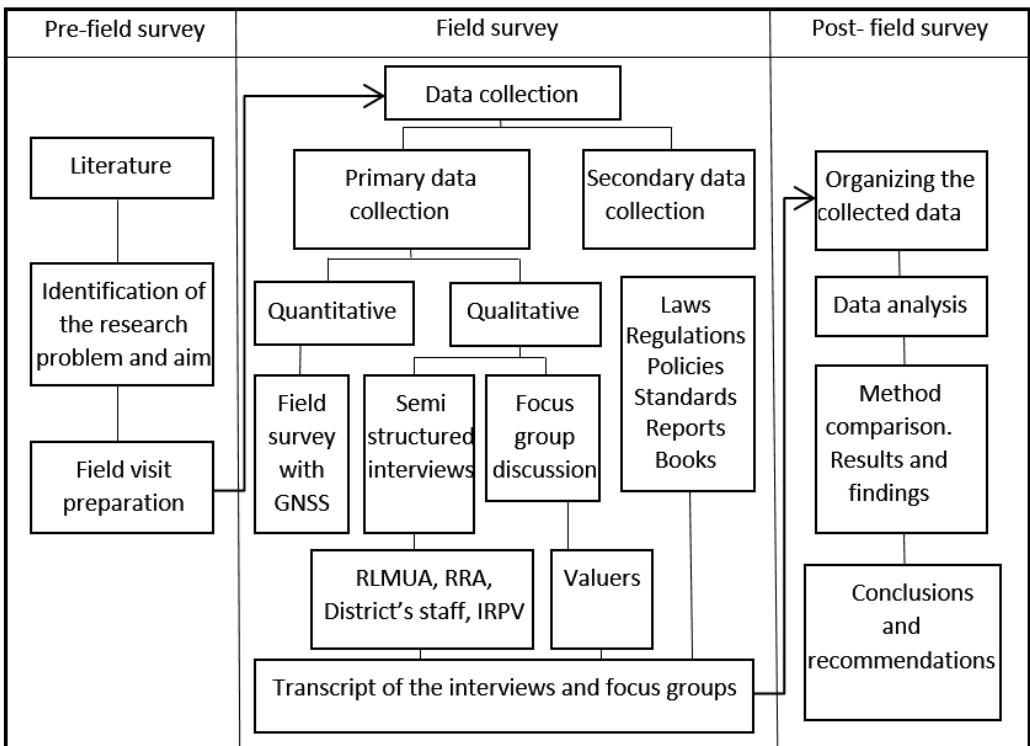

**Figure 4.** Research workflow.

**Table 2.** Number of respondents by institution.

| Institutions | Category | Number of People |
|---|---|---|
| RLMUA former RNRA | | 2 |
| RRA | Central government | 3 |
| Ministry of Infrastructure | | 1 |
| District staff | Local government | 3 |
| RCMRD | Regional | 1 |
| IRPV | Private institution (private valuers) | 3 |
| | Total | 13 |

Further data was collected using focus group discussions, involving thirteen people with a background and interest in valuation [37]. The topic of the discussion was "Comparison of remote sensing techniques for valuation for taxation in Rwanda". The aim of the interviews and the focus group discussion was to collect individual and group information on perceptions of the current property valuation for taxation purposes and to assess the advantages of the different remote sensing approaches for property valuation for taxation. Open-ended questions were used [38] as a quantitative approach for collecting numerical data. The qualitative method was used to understand how the current property valuation for taxation system works in Rwanda and who are the key players and their specific roles. The perceptions of the stakeholders, especially government institutions, on using the remote sensing data for property valuation for taxation purposes was also investigated.

*2.3. Methods for Data Processing and Analysis*

In terms of analysis, for the RS dataset comparison, that is, the assessment of the usefulness of the different remote sensing techniques, four properties in the study area were selected based on the criteria, which are shown in Table 3, inspired by FFPLA elements. Moreover, as the acquisition years of the three datasets are different (2008, 2013, and 2015), the properties have been selected where there is no change during these years in terms of their area. This selection was especially needed for the accuracy assessment.

**Table 3.** Criteria of sampled and surveyed properties.

| Sampled Property | Criteria |
| --- | --- |
| Property 1 | Accessibility<br>Clearly visible boundary<br>Developed land (Building improvements)<br>Visibility of the building footprints<br>Comparison based on all RS data |
| Property 2 | Accessibility |
| Property 3 | Comparison based on all RS data |
| Property 4 | Developed land (Building improvements)<br>Visibility of the building footprints<br>Comparison based on all RS data |

An analysis of the qualitative data guided by FFPLA elements' [1] text-based analysis is used, dividing the factors into the following themes: (1) factors affecting or influencing the property value; (2) characteristics of generated orthophotos from remote sensing data; (3) time/availability of the platform; and (4) cost of acquiring the data, including the cost of hiring the platform. ArcGIS software was used for the quantitative assessment and the comparison of the parcel areas, location visualization of the subject property and neighborhood, and the coordinate comparison. The interviews and focus group discussions from participants were transcribed and analyzed using ATLAS.Ti 8.0 based on the developed themes. Thematic analysis based on the literature review on the factors was used to examine the collected data.

## 3. Results

In this section, the results are presented in separate sections following the idea of the legal, governmental, and technical framework introduced in FFPLA. In the first section, policies, laws, standards, and types of property taxes are described. This is followed by the data requirements and methods for data collection and analysis. Afterwards, the existing limitations for using RS data and considerations related to its application and comparison with the newly proposed UAV method are shown.

### 3.1. Policies, Laws, Standards and Types of Properties in Rwanda

The real property valuation profession in Rwanda is regulated by law N° 17/2010 [39]. It specifies the structure of the Institute of Real Property Valuers (IRPV) and defines their responsibilities. It describes the methods currently used for data analysis, but it does not specify any methods for data collection for property valuation [39]. Before 2010, property valuation was carried out mostly by civil engineers who had some training and experience in property valuation methods [40].

During the interviews with the IRPV officials, two of the three respondents stated that: "the valuation standard in Rwanda work as a practicing guide".

However, there is confusion in understanding and applying the standard and law at a local level. Therefore, international valuation standards are used. It is the responsibility of IRPV to develop the valuation standard, and it requires the approval of the property valuation regulatory council before using it.

The two forms of tenure in Rwanda are full ownership (freehold title) and long leasehold title. About 98% of Rwandan land is held under leasehold and 2% under freehold. Rwanda's land law requires landowners with a freehold title to pay property tax while emphyteutic leaseholders pay lease fees. This means that only 2% of Rwandan land is taxable. The Rwandan National Land Policy requires that "land transactions and land taxation be included in land administration as elements of land development" [41].

The current property tax (fixed asset tax) is amended by law N° 75/2018 of 7 September 2018, which determines the sources of revenue and property of decentralized entities. Additionally presidential order N° 25/01 of 9 July 2012 establishes the list of

fees and other charges imposed by decentralized entities and determines their thresholds. The current laws related to property tax in Rwanda are categorized into three different taxes as specified by the taxation law and confirmed by the Rwanda Revenue Authority (RRA), which was confirmed by districts officials interviewed during fieldwork, including fixed asset tax, land lease fees, and rental income [29]. During the interviews, respondents highlighted that "fixed asset tax or property tax requires a valuation report and its taxes are based on the market value whereby the tax value is equal to 1/1000 of the open market value of the property".

Ministerial order N° 005/12/10/TC of 22 June 2012 determines the modalities for the implementation of law N° 59/2011 of 31 December 2011. Ministerial order N° 005/12/10/TC of 22 June 2012 determines the modalities for the implementation of taxation law. It specifies the process and suggests the steps to be followed by the taxpayer for all types of taxes. The steps and key actors involved in property tax (fixed asset tax) are visualized in Figure 5 below. These include RRA, districts, and taxpayers, Rwanda Land Management and Use Authority (RLMUA), valuers, Institute of Real Property Valuers (IRPV), and Rwanda Development Board (RDB).

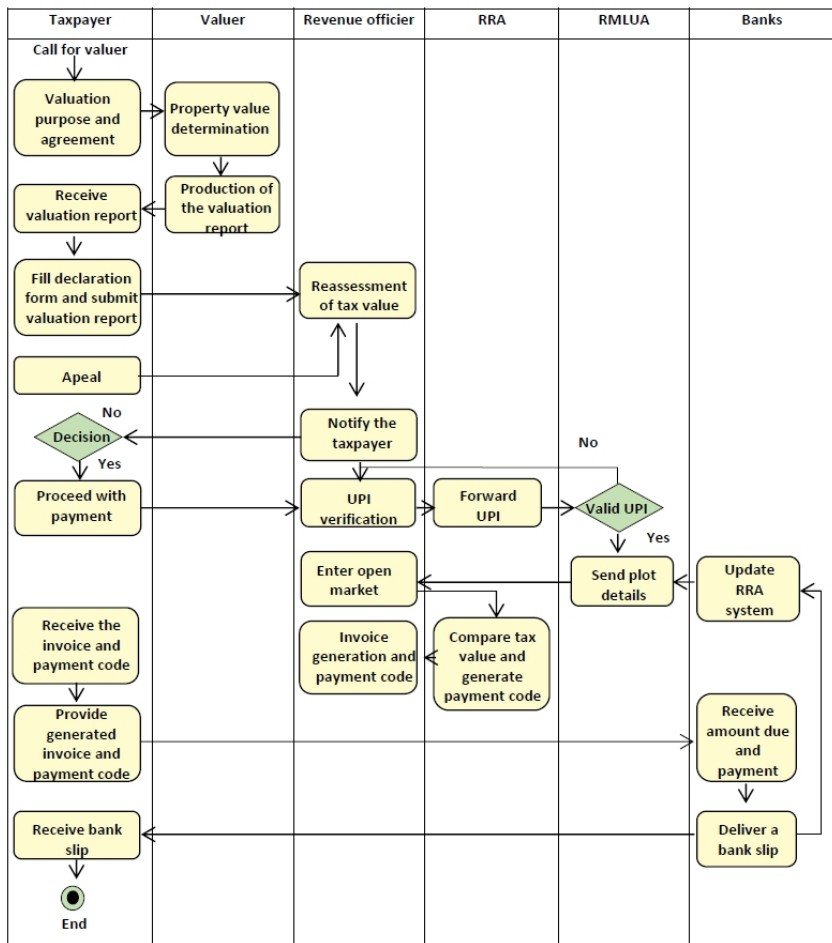

**Figure 5.** Property tax system in Rwanda.

Three types of property taxes are levied on property in Rwanda. They are lease fees, rental income taxes, and fixed asset taxes (derived from the interview with district and RRA officials). The difference between these three types of taxes is based on the land tenure type. Land lease fees are the tax levied from leaseholders; freeholders pay a fixed asset tax, and rental income tax can be levied on both land tenure regimes. All these taxes are levied on an annual basis. The only type of tax that requires a valuation report is fixed asset tax as it is based on open market value.

### 3.1.1. Land Lease Fees

Land lease fees are paid annually. The method used to determine the tax value is based on the rate per square meter or hectare. The rate is determined by the district council depending on the infrastructure and development in an area. The presidential order determining the list of fees and other charges levied by decentralized entities and determining the thresholds in article 9 states that: "Any person owning land and holding a land lease certificate issued by the competent organization shall pay an annual land lease fee based on the square meter or hectare". The same article declares that it is the responsibility of the district council to determine the fees to be paid annually based on the available infrastructure in the area where the land is located and its use. The thresholds of land lease fee rates are arranged from 30 Rwf (around 0.03 euro) to 80 Rwf (0.08 euro) per square meter and (4000 Rwf) per hectare levied from agriculture and livestock land with more than two hectares.

### 3.1.2. Rental Income Taxes

Rental income tax is a tax imposed on individuals who earn income from the rented immovable property. Taxes are paid annually based on rental fees the landowners collect from their tenants, and this tax is also paid in terms of percentages of generated income from the property after deduction of expenses. Taxation law article 50 states that the expenses should be 50% of the generated income per year. The tax rate ranges from 0–30%, depending on the generated income. The more income generated, the higher the rate applied.

### 3.1.3. Fixed Asset Taxes

Fixed asset tax, also called "property tax" is a tax imposed on immovable property with a freehold title. Through interviews from district and RRA officials, they highlighted that: "property tax requires a valuation report as it is based on the open market value that is why they require an expert in valuation to determine the open market value". Fixed asset tax value is normally based on the value of the property. The rate that is applied to the value is fixed at a thousandth (1/1000) of the taxable value per year. During fieldwork, interviewees underlined that: "they are still facing a problem of taxpayers who declare the property and hide some information related to their property for instance when they have a number of buildings within a few parcels, because the purpose is for taxation they undermine the value and report only one building". However, a mechanism of monitoring changes on the ground is needed so that taxpayers do not hide information that can be captured easily.

Fixed asset tax value can be updated once every four years as stipulated by taxation law article 15. However, if improvements or changes are made to the property, the taxpayer must file an updated valuation report (new self-assessed tax) and fill in a new tax declaration or assessment notice [29]. Interviewees highlighted that: "if a property is residentially used, the fixed asset tax value should be determined after deducting the amount equal to three million (≈3000 EUR) on its market value". This is specified in article 18, point number 8 of the taxation law, on tax exempted properties.

### 3.2. Data Requirements for Property Valuation for Taxation in Rwanda

The required data for property valuation, based on the current approaches for taxation purposes, depends on the type of property and its use. The most important requirement is the land ownership titles (lease or freeholds). A land title is the one that shows the basis upon which the land is held (whether it is under a leasehold or freehold title). Taxpayers with freehold titles require valuation reports as tax basis calculations. The necessary information required to value the property for taxation purposes, as said by participants, includes: "land certificate ownerships, built-up area, technical conditions of the property, construction materials, infrastructure attached to it, land use".

Most of the information is collected from the field, and others are retrieved from the land title (LAIS database) (Table 4).

**Table 4.** Required data and source availability.

| Required Data | Fieldwork | Land Title LAIS | Masterplan | Google Earth | Valuers | Estate Agencies |
|---|---|---|---|---|---|---|
| Parcel area | ✔ | ✔ | | | | |
| Current land use (on title) | ✔ | ✔ | | | | |
| Built-up area | ✔ | | | | | |
| Construction materials | ✔ | | | | | |
| External works | ✔ | | | ✔ | | |
| Status of the property | ✔ | | | | | |
| Infrastructure attached | | | | ✔ | | |
| Planned land use | | ✔ | ✔ | | | |
| Sales comparable | ✔ | ✔ | | | ✔ | ✔ |
| Location | ✔ | ✔ | ✔ | ✔ | | |

The taxes for leaseholders do not require a valuation report. All the required data for tax calculation is based on the Unique Parcel Identifier (UPI), which is obtained from the land lease title and district council resolution. The tax value is based on a rate per m$^2$ rather than on the open market value. Thus, fixed asset tax (property tax) is based on the open market value and requires: "sales contract value or certificate of valuation by the certified valuer to fix the open market value".

The most highly trusted sources of the required data, as discovered during the fieldwork, include RLMUA, estate agencies, and valuers. However, RLMUA data were found to be incomplete, inaccurate, and outdated in some cases. Focus group participants confirmed this, stating that: "the land use on the title differs from land use on the ground, the size of the parcel on the ground differs from those in the system or title, while the number of houses within a parcel is missing from the Rwanda land cadastre". It is still a challenge in Rwanda to get all the required data for the valuers to support their value assessment. During the focus group discussion, the valuers highlighted that: "valuation depends on the available data, purposes of valuation, and use of the property".

For instance, for the income-generating properties, valuers need to look at the book account and see the income the property is generating. The property value is calculated by capitalizing of the future income from that property. Thus, many valuers depend on the property used to assess its value. This was also confirmed by participants from the focus group discussion, who claimed that: "there is a lack of data to use the appropriate methods to evaluate the property according to its type, which results in the replacement cost method being used more compared to other methods of property valuation".

### 3.3. Data Collection and Analysis Methods for Property Valuation for Taxation in Rwanda

The current data collection methods consist of field visit (inspection of the property), inspection of Google Earth/Maps, and consideration of the masterplans. During a field visit and inspection of the site, the valuer has to use different methods and tools for data collection for property value determination such as: "Tape measurement, digital camera, and handheld GPS and laser distance meter". Whilst the tape measurement is usually used for buildings and improvements, the parcel areas are obtained from the land title. Throughout focus group discussions, participants underlined that the tape measurements are used to: "measure the size of the houses, gross floor area, improvement, and other external works such as drainage system, parking, gardening" to complete the data from land titles (LAIS database). A digital camera is used to take pictures to: "ensure that the property exists at the date of inspection and to have a better visualization of their physical appearances, the materials and technical conditions of the subject property to be valued". This was highlighted by the valuers during the focus group discussion. Acquired images are used in property valuation as a source of information and prove their existence

at the date of valuation. Currently, Rwandan valuers use handheld GPS to verify the location and measure the size of the property to be valued. During the interview and focus group discussions, the respondents said that: "even if handheld GPS is being used in data collection, its accuracy of 3 m, is not good as it should be".

A laser distance meter is used to measure the internal and external parts of the buildings. According to the users, this tool is more precise, and measurements can be done faster than with tape. Google Earth is used for location, verification, and visualization of the existence of the property at that parcel and confirms their existence on the ground at the date of valuation. In addition, the masterplan is one of the districts' planning outputs, and it is prepared for a period of 10 to 20 years. It is used mainly in urban areas to help valuers to know the allowed land use and zoning of a particular area.

The current technical steps involved in carrying out property valuation for taxation purposes in Rwanda are shown in Figure 6. The process starts with a call from the client (taxpayer) or tender bid advertisement of institutions (Private, Public, or NGOs). During this phase, an agreement for a meeting is scheduled (oral or written). After the agreement is done, the next step is to conduct a field inspection for fieldwork. It has to be conducted by a competent valuer in the presence of the landowner. If the valuation is for developed land, measurements of the buildings and notes on improvements within the compound of the parcel are required, while for undeveloped land the information provided on the land title is used, unless there are unreported changes on the land itself. During the fieldwork, the valuer has to collect the required data for valuation, and some pictures are taken for further analysis. The office work consists of analyzing the collected data and preparing the report. As a result, the validated valuation report is provided to the taxpayer. Updating of the valuation report also follows the same steps, unless there are no changes made to the property. However, the valuer should be well informed that there are no changes that happened in the last four years.

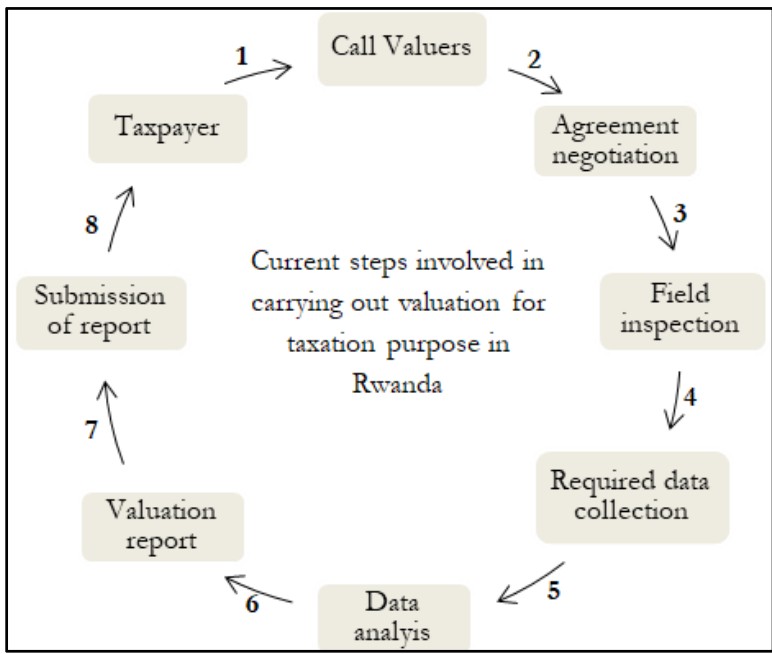

**Figure 6.** Steps of the current valuation system in Rwanda.

### 3.4. Existing Use of RS Data in Rwanda

Remote sensing data has been used in Rwanda since 2008. During the interviews, we found that in public institutions, such as RLMUA, RS data were used mainly in the creation of the digital cadastral maps. The interviewees highlighted that: "the Rwanda land cadastre was built based on the high-resolution orthophotos captured in 2008 using aircraft and satellite images. Satellite images in particular were used in the north-western part of

the country. Due to the topography of that area, it was not possible to cover that area using aircraft". The generated orthophotos were very useful in identifying entire properties. Given that the images of 2008 are becoming outdated, the valuers often overlay them on top of the latest available satellite images. RLMUA, as a government institution, highlighted the importance of high spatial resolution. Currently, the government of Rwanda has signed the "Memorandum of Understanding with the government of Gabon under which both governments will be sharing spatial information and expertise in land registration". The government of Gabon will be providing satellite images as they have satellite centers, while the government of Rwanda will be providing expertise in land registration using a fit-for-purpose approach. The interviewee shared that "generally, the usability of remote sensing data in property valuation for taxation purposes is very low, for instance, the most used images are those captured during fieldwork and those images that can be downloaded from Google Earth, which often have a low spatial resolution".

### 3.5. Identified Limitations for Using RS Data for Property Valuation for Taxation in Rwanda

The general findings from interviews and focus group discussions are that the biggest challenges are a lack of data due to a lack of funds and skilled professionals to use them. In Rwanda, there "are no specific laws or regulations governing the use of remote sensing tools". This was highlighted during the interview session. The use of remote sensing tools is governed by the law regulating Civil Aviation in Rwanda. The availability of the platform is another challenge emphasized by the interviewees because they have to be ordered outside of the country. During the interview discussion with the Ministry of Infrastructure officials, they highlighted that "only one company has been registered and has the right to fly UAVs, but, for other remote sensing techniques, they need to hire international companies to carry out the photogrammetric acquisition and processing".

During the interviews, the applicability of different remote sensing images and the possible challenges associated with them were discussed, with examples shown below in Figures 7 and 8. Figure 7 shows the property visualization of sampled property number four in the different remote sensing datasets at the same scale (1:700). The results show that the features on the UAVs-orthophoto in Figure 7a are better visible than the satellite and aerial orthophoto, as shown in Figure 7b,c. The construction materials such as roof cover and pavement or external structures are also much clearer in the UAV image.

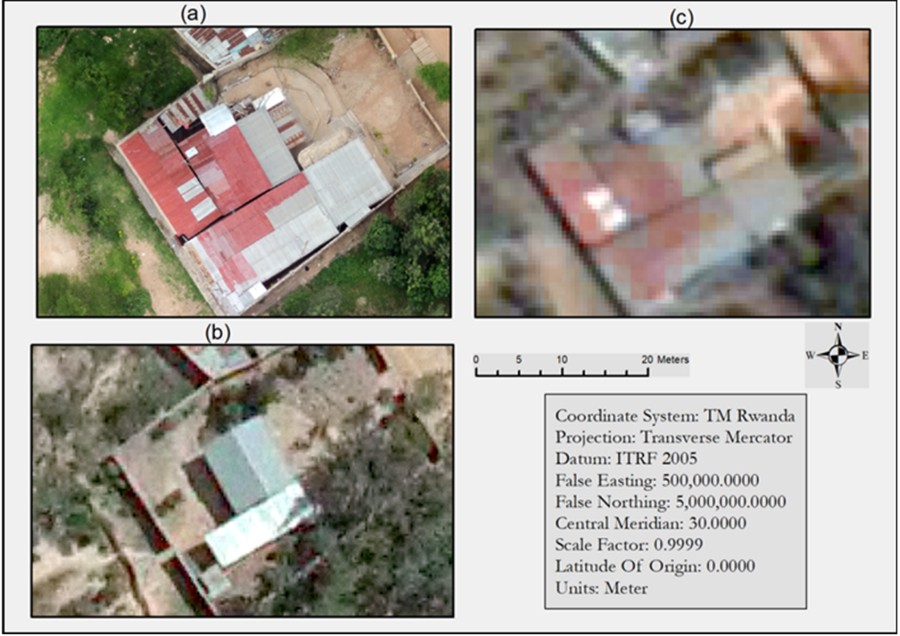

**Figure 7.** Property visualization from three used datasets: (**a**) from UAV, (**b**) from aerial orthophoto, and (**c**) from Satellite-Worldview 2.

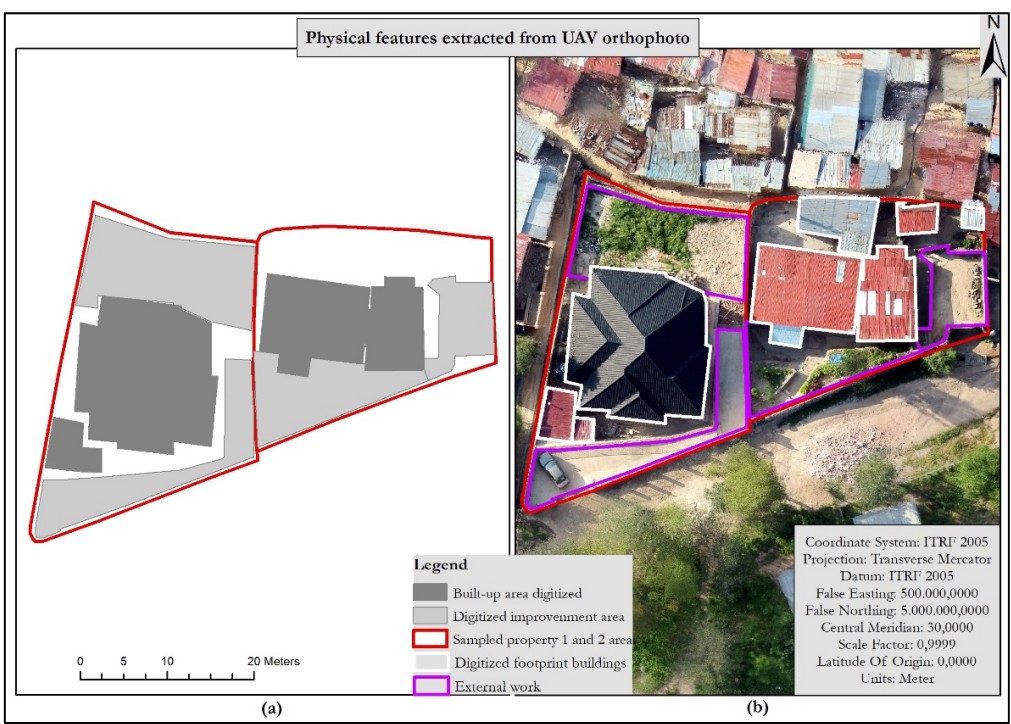

**Figure 8.** Features extracted from UAV orthophoto (**a**). Overlaid features on the UAV image (**b**).

Therefore, among these data sources, UAVs definitely outperform the others. UAV images seem suitable for determining the physical and locational characteristics of the property and its improvements. Based on the orientation of the camera (inclined or oblique) during image acquisition, information on the technical condition of the property, such as facades of the buildings and construction materials, can be obtained. The visualization of such information enables valuers to remotely evaluate the physical obsolescence of the property with high precision. Apart from information about the taxable property, the quantitative data such as parcel area, built-up area, perimeter, and distance to public facilities can be determined from the generated orthophoto (Table 5 and Figure 8).

**Table 5.** Physical characteristics of property identifiable on UAV-orthophoto for sampled property 1 and 2.

| Physical Factors | | | | | | Location Factors | | | | |
|---|---|---|---|---|---|---|---|---|---|---|
| Property (Land or Buildings) | Parcel Area (m²) | Built-Up Area (m²) | Improvement (m²) | Shape | Type | Neighborhood | Accessibility | Land Use | Environment | Utilities |
| Property 1 | 708 | 275 | 282 | regular | built | yes | yes | - | yes | yes |
| Property 2 | 557 | 247 | 175 | regular | built | yes | yes | - | yes | yes |

### 3.6. Consideration of the Potential for RS Data Application to Property Valuation for Taxation

The assessment of the different information that can be retrieved from remote sensing data is done based on the factors derived from the literature, visualized in Figure 1 and adapted for this study. These factors are classified into two main categories: internal and external factors. Internal factors include parcel size, built-up area; improvement; shape, and type of subject property, whereas external factors include accessibility, neighborhood, land use, environment, and utilities.

#### 3.6.1. Internal Factors

The **parcel size** is used to determine the land value of either the developed or undeveloped parcels. This serves as the tax base for both leaseholders and freeholders. Moreover, for the land lease fee to be paid by the taxpayer, parcel size plays a very important role because the bigger the parcel, the more lease fee is paid if the location of the parcels are in

the same area whereby the rate is the same. Parcel area serves as a land lease fee basis as the method of determining the lease fee is the rate per square meter. Additionally, for the property tax where the parcel is not developed, the open market value is determined based on the rate per square meter. However, the applied rate for lease fees and market value differs. The last one must be determined by the private valuers on the basis of recent sales, location, and infrastructure within the area and public facilities surrounding the area.

The valuers have the right only to report if the size on the title does not reflect the actual size on the ground. The authority who is intending to use the report has to decide on the above-reported errors if corrections should be made. The valuers must prove the mismatch of the areas. This can be completed using remotely sensed data. As shown in Figure 9, the parcel area for Property 2 manually digitized from UAV-orthophoto (b) is closer to the area measured with GNSS (ground truth) compared to manually digitized from the satellite image (c) and existing area in LAIS database, which is based on the orthophoto from aerial images from 2008 (a).

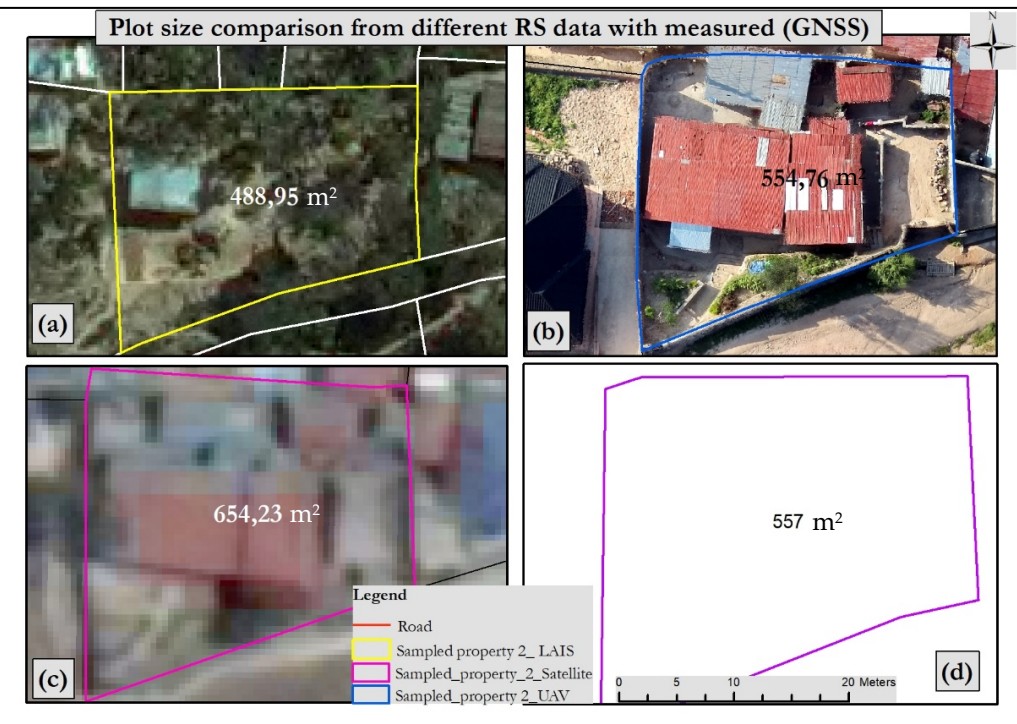

**Figure 9.** Plot size comparison of sampled property 2 from different remote sensing data: (**a**) existing parcel area in the LAIS database (based on orthophoto from aerial images 2008), (**b**) manually digitized from UAV-orthophoto, (**c**) manually digitized from satellite and (**d**) measured with GNSS during the field visit.

The **built-up area** is one of the most relevant factors for the property tax system. Property tax is imposed based on market value, and this includes the value of the land, the building, and any improvements. Therefore, precise area calculations are needed. Moreover, the size in the vertical dimension also should not be forgotten. In the built-up-areas, the numbers of floors can be visible if there are oblique images (Table 6). The participants from the focus group underlined that "due to the lack of data, especially related to buildings and improvements, the fieldwork measurement and image acquisitions are compulsory as the current property tax regime is based on open market value, and it depends on the determined information from the field".

**Table 6.** Built-up and parcel area manually digitized from the UAV orthophoto of sample properties 3 and 4.

| Property (Land or Buildings) | Parcel Area (m²) | Built-Up Area (m²) | Improvement (m²) | Shape of Parcel | Type | Neighborhood | Accessibility | Land Use | Environment | Utilities |
|---|---|---|---|---|---|---|---|---|---|---|
| | **Physical Factors** | | | | | **Location Factors** | | | | |
| Property 3 | 399 | 263 | - | regular | built | yes | yes | - | yes | yes |
| Property 4 | 2471 | 417 | 278 | regular | built | yes | yes | - | yes | yes |

The purpose of acquired images is to show the existence of the property and to visualize technical conditions and the physical appearance of the property (Figure 10).

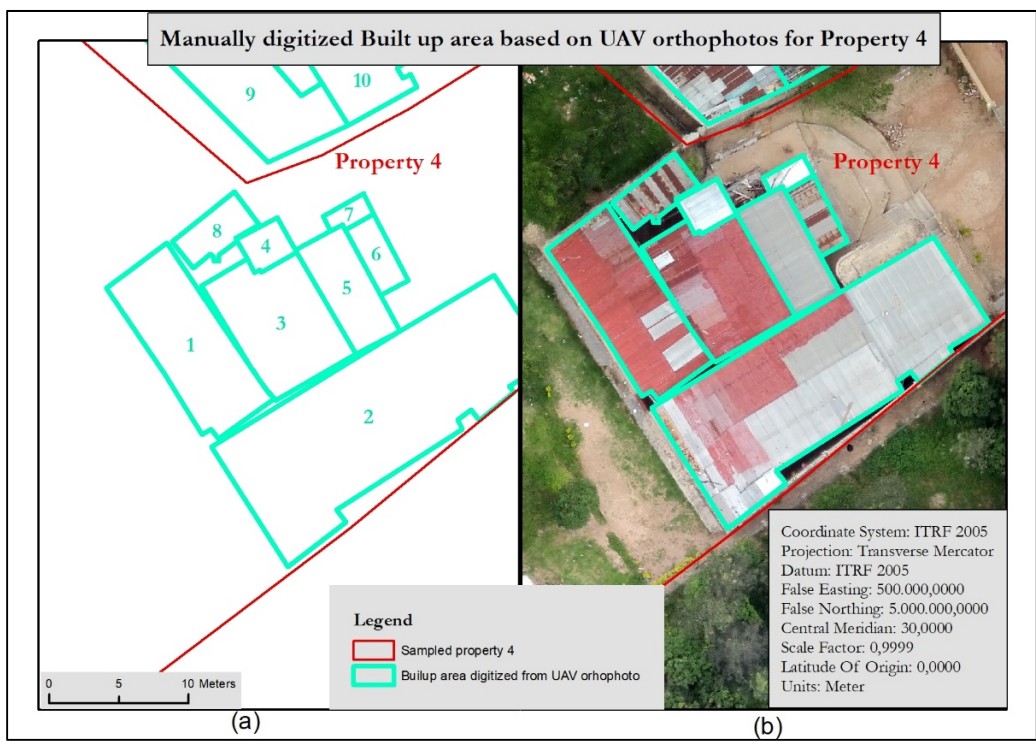

**Figure 10.** Digitized building footprints from UAV-orthophoto of property 4 (**a**). Footprint outlines overlaid on the UAV-orthophoto (**b**).

**Improvement** refers to the level of external work done to increase the value of the property; this includes property maintenance, concrete parking, gardening, sewage drainage system, water tank, and others. These features are also part of the market value of the property to be taxed and need to be assessed and considered during data collection for property valuation for taxation purposes. Therefore, the size of these external structures is required. However, they are not recorded in the current cadastre, but, can be extracted from images as shown in Figure 11. Moreover, if terrestrial images can be obtained, they can be used as supplementary materials, and in combination with UAV images, quite detailed information can be provided for valuation purposes.

**Shape** refers to the spatial form of a parcel, whether it is regular (perpendicular lines) or irregular (polygons with curves). The shape does not affect the value of the land and property directly, whereas it affects the improvement and design that can be put up in that parcel.

The **type** refers to whether the parcel of land is built or unbuilt. However, this goes hand in hand with land use, and development conditions that can be filled by specifying the number of buildings requires a construction coverage ratio and allows for a number of floors on an individual parcel.

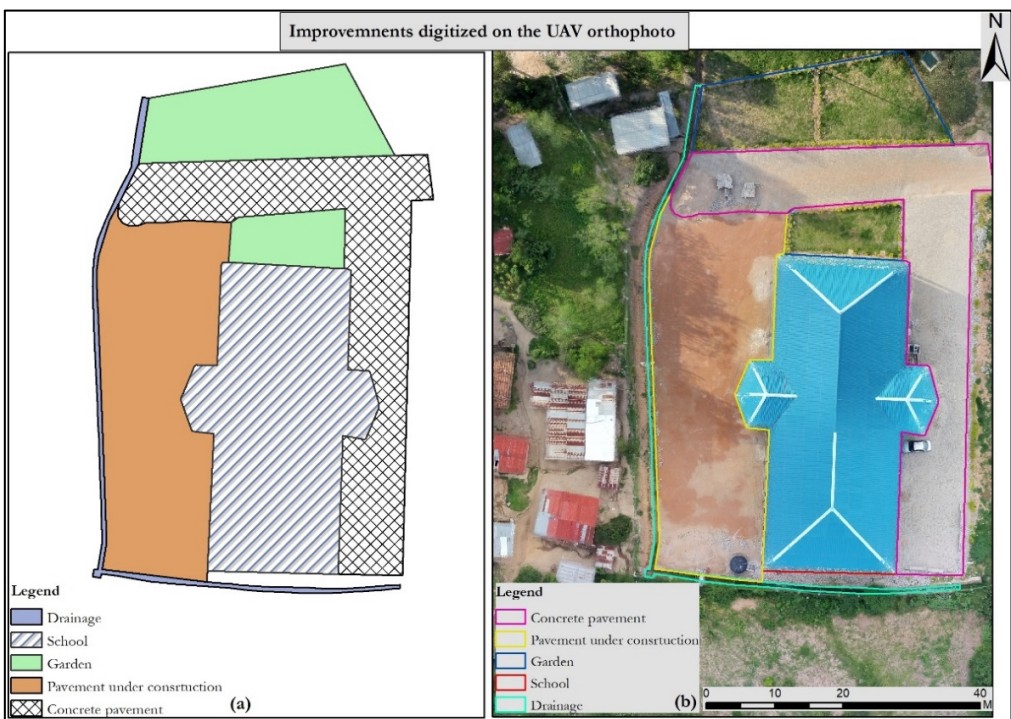

**Figure 11.** New features (improvements) digitized from UAV-orthophoto of property 4 (**a**). Outlines overlaid on the UAV-orthophoto (**b**).

3.6.2. External Factors—Location Factors

As shown in Figure 12, location factors such as neighborhood, accessibility (roads) of the property, public facilities, utilities, and infrastructures, all affect property value. The neighborhood is a crucial factor in the specific surroundings that affect the value of the property. The surrounding features of the property are determined by the buffer around that property in a particular area. Throughout fieldwork, the participants from the focus group discussions underscored that "location is the most important factor to be considered compared to the physical factors". They ranked the neighborhood factor as the first factor influencing the property value in a specific area. The levels of surrounding development to the property affect its value. These developments include transport facilities, retail outlets, service outlets, and public facilities, such as schools, markets, health centers, parks, churches, hospitals, manufacturers, and others. For instance, a property that stands out as being too different from the others will also differ in price, even if it is in the same neighborhood (Figure 12).

**Accessibility** is measured by how accessible the property is. During the discussion with valuers, they highlighted that "accessibility is more related to the road infrastructure and defines whether the property is accessible by primary roads or districts roads, water pipes, fiber optic internet, and electricity". For instance, if the property is directly attached to tarmac roads, as shown in Figure 13 for properties 3 and 4, it is more costly than a property located on a marram (**clayey/sand**) road in the same area. Accessibility was ranked as the third factor influencing the value of a specific property after neighborhood and construction materials.

**Land-use zoning** can either be the current use of the land or intended use as specified by the master plan. In Rwanda, the land use master plan for the entire country was developed in 2011. The district development plan was developed by referring to the developed national land-use masterplan. Property valuation considers the highest or best use of the land as it is being used as intended or planned to be used depending on the surrounding developments in those areas. Results from the focus group discussion

with valuers in Rwanda underlined that "the highest and best use of the land is tangible potential, officially permitted, most economically feasible, and outstandingly profitable".

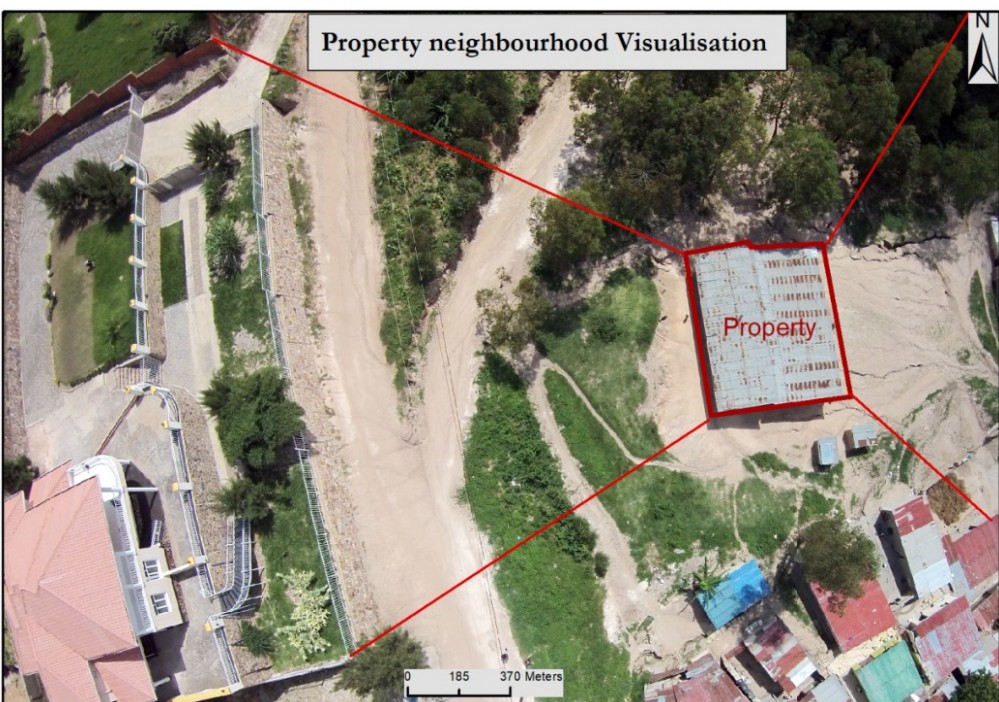

**Figure 12.** Neighborhood visualization of the property based on the UAV orthophoto.

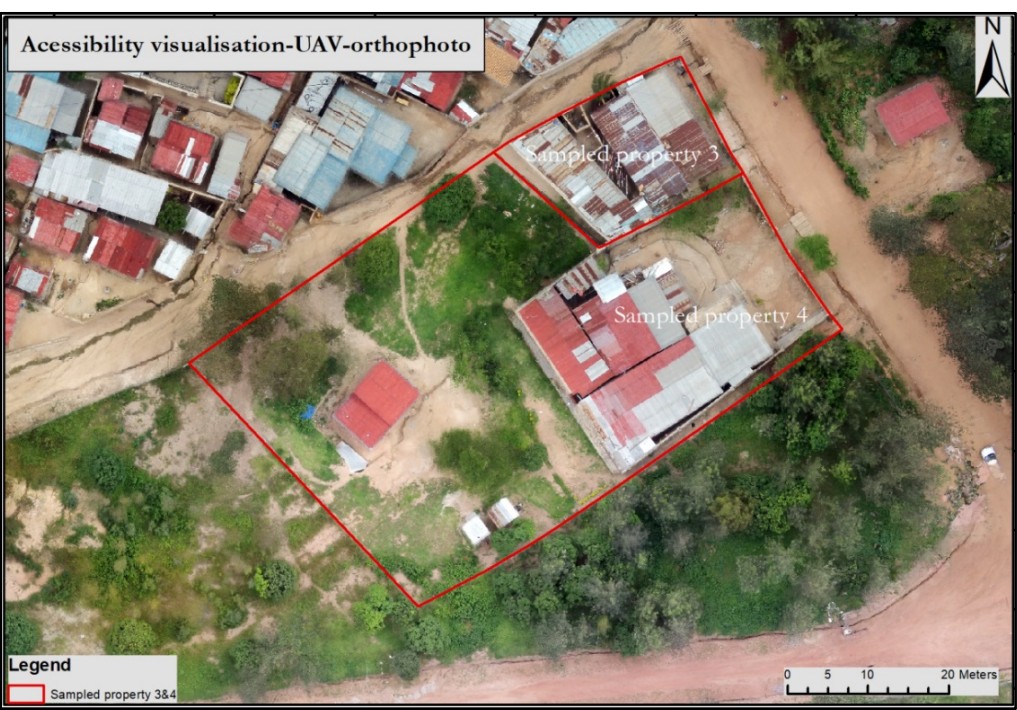

**Figure 13.** Accessibility visualization of sampled properties 3 and 4 from UAV-orthophoto.

For valuation purposes, it is important to examine and put into consideration the issues of zoning and its changes. Additionally, land use planning laws need to be considered. For example, the development project must fulfill the requirements of the intended land use of the area by constructing the proposed structures on the specific parcel. Throughout our discussion, the valuers thought that "the masterplan of all districts should be online and

open to the public as the Kigali master plan 2013 is user friendly and is used as a method of land use checking and to locate subject properties".

Figure 14 shows the planned land use of sample property 1. All permitted and prohibited constructions are specified in the zoning guidelines, which can be downloaded from the master plan (accessible online). The current master plan was developed based on the orthophotos from aerial images taken in 2008, and this is being used in the property valuation profession due to the fact that valuers use it as a credible source of information.

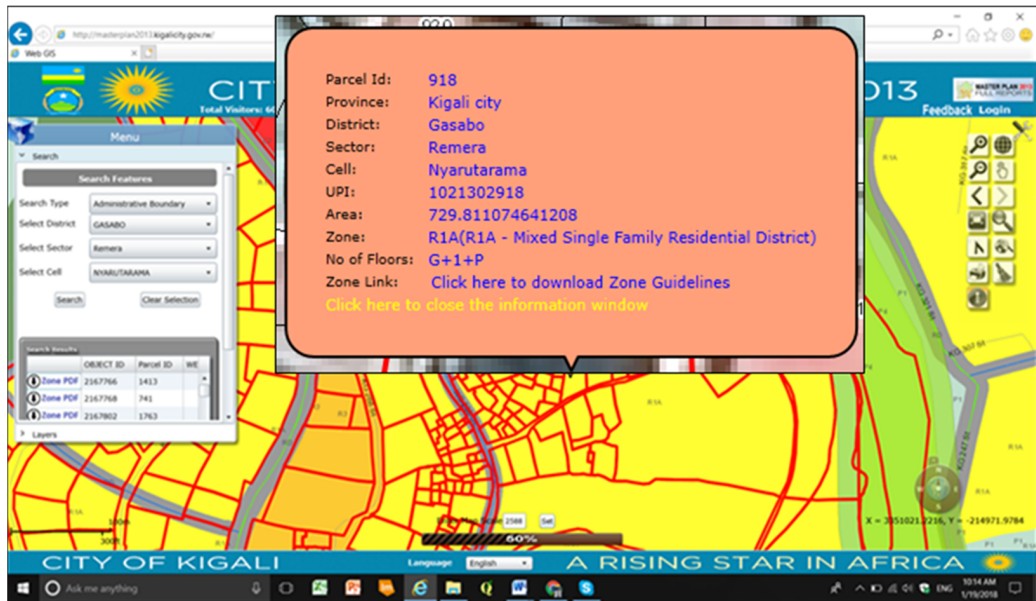

**Figure 14.** Land use of sample property 4 via the Kigali masterplan 2013 (Source: [42]).

The **environment** in this research refers to a geographical condition, a specific area where the property is located. The interviewee told us that "the area disposed to the effects of natural phenomena, such as flooding, high winds, and earthquakes, among others, are poor choices when buying property". Property that is located near wetlands (informal settlement areas) is worth less than that located in an approachable environment. Remote sensing data can be useful to analyze and visualize the prone area compared to the location of the property.

The **utilities** in this research refer to the services or features that are connected to the properties, such as water pipes, electricity, gas, sewerage drainage, and other facilities. With remote sensing data, utility features can be distinguished from other features. With UAV-orthophotos in particular, more accurate and precise information on the ground can be extracted compared to satellite images and orthophotos from aerial images.

### 3.7. Comparison of the Proposed New UAV Based Method for Property Valuation for Taxation

Based on the abovementioned factors, UAV images outperform the other remote sensing images. Therefore, during the interviews and focus group discussion, we assessed their applicability as a newly proposed method to be used for valuation for taxation in Rwanda. The elements for the assessment that we used were accuracy, completeness, up-to-datedness, cost, and availability of the platform.

Evaluation of the spatial accuracy was also done through a comparison of the area of property 1, 2, 3, and 4 digitized over the three remote sensing images and compared with the reference data measured with GNSS. As shown in Figure 15, since the resolution of the UAV image is much higher than the other images, the digitized area is much closer to the reference data. It should be mentioned that due to the coarse resolution, property 4 was not clearly visible for digitization on the satellite image. Therefore, it is missing in the figure below.

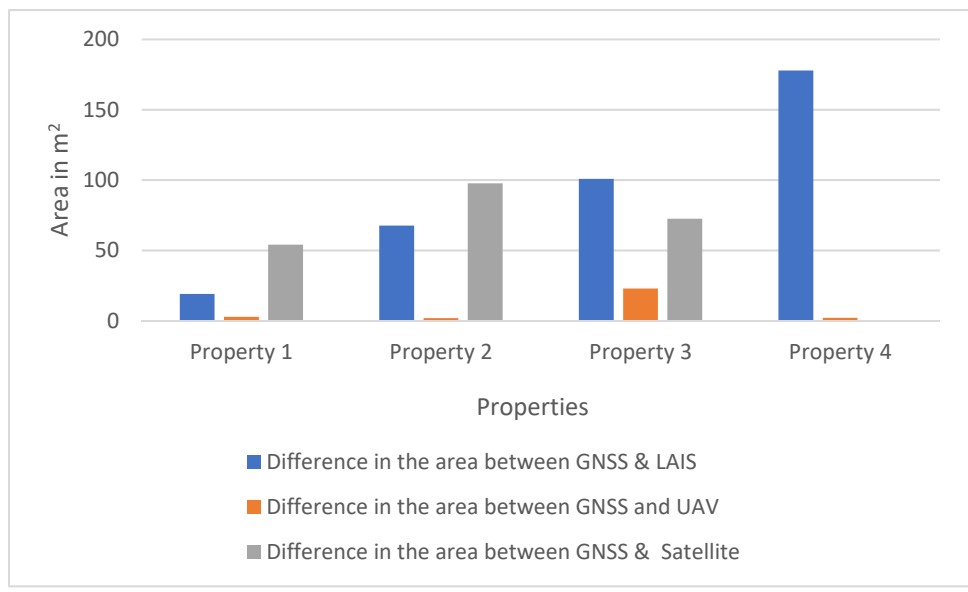

**Figure 15.** Area comparison between the measure of GNSS of four properties and polygons from the existing LAIS database based on digitization over an aerial image taken in 2008 (blue), UAV orthophotos (orange), and satellite images (grey).

Evaluation of **completeness** was based on the answers from both semi-structured interviews and focus group discussions with IRPV officials, valuers, and RRA officials. It was done by showing the features extracted from UAV-orthophoto and the other RS data, compared with the required data for property valuation. Findings from focus group discussions revealed that "the interior part of the property cannot be captured using remote sensing data and construction materials can be obtained in ascertained condition". Regarding the completeness of data, RLMUA interviewee and focus group discussion participants highlighted that "current information provided for property tax (fixed asset tax) is not enough (incomplete) and the information related to the building improvements within the compound of the parcel are not recorded in the LAIS database". Therefore, remotely sensed data, especially high-resolution ones such as UAV images, can be used to compensate for this gap. The only missing element that the participants shared is that RS data cannot provide information about the inside conditions of the properties.

The degree to which remote sensing data provides **updated** information was assessed via literature review, interviews, and focus group discussions data from RLMUA, IRPV, RRA, districts, and valuers. After the aerial data acquisition in 2008, the cadastral data has not been updated. Rwanda authorities have planned to update the existing spatial data in five years for urban areas and ten years for rural areas; however, satellite images from Google Earth and sporadic field measurements are still currently used where needed. During the interviews, valuers said that RLMUA, as an institution in charge of the spatial information related to land management and use, should find the approach for updating spatial information so that the data from the database reflects the reality on the ground. In relation to property taxation, the results from interviews with RRA and district officials concluded that the "classical approach of property to property self-declaration from land-holders is being used for updating their data". This traditional approach is time-consuming and costly, and interviewees stated that "remote sensing techniques can be more useful to monitor and keep the spatial information regarding the changes on the ground up-to-date, which can save time compared to the traditional method".

The **cost** assessment is based on the previous research done in Rwanda and abroad on the usability of UAVs compared to the valuation fees in Rwanda. IRPV [43] has set valuation fees. They classify the properties into different categories. The categories include factors such as type (land or building); use (residential, commercial, and industrial), and

location (urban, peri-urban, and rural area). What costs need to be considered for the tasks related to the current procedure and the one based on UAVs are shown in Figure 16.

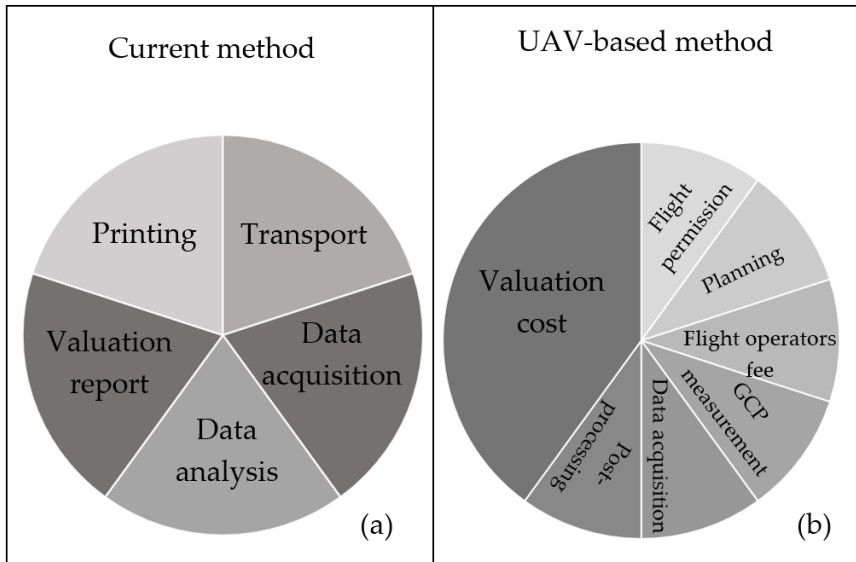

**Figure 16.** Cost assessment of the current property valuation method (**a**) and the proposed UAV-based method (**b**).

As shown in Figure 16, for the current valuation procedure, costs should be planned for transport, data collection, data analysis, valuation report preparation, and printing. According to [43], the average price of the charges for property valuation in an urban area is around 1050 Rwf per m$^2$. For the UAV-based method, the costs were calculated based on information provided by the Charis Company, the only company registered and allowed to fly in Rwandan territory. Based on their experience, a UAV approach should consider costs for obtaining flight permission, pre-planning activities, operator fees, ground control points (GCP) measurement, data acquisition, and post-processing. According to these data, the estimated price is $1100 per five hectares, which is equivalent to 18.95 Rwf per m$^2$. The price was converted into Rwf francs for better comparison. The online exchange was used as the exchange rate at the time was 861.42 Rwf /dollar. After the data acquisition, costs for valuation report preparation, analysis, and printing need to be included.

In terms of **availability of the platform**, during the interview discussion an official from the Ministry of Infrastructure shared that currently "only one company is registered and has the right to fly UAVs".

## 4. Discussion

This section is organized as follows. First, the overall potential of remotely sensed data in land valuation for taxation in Rwands is discussed. Second, a qualitative cost/benefit analysis of the proposed UAV methods for property valuation for taxation. Third, the comparison of, and discussion on, the current procedures and the proposed new methods are shown.

First, on overall potential for RS utilization in land valuation for taxation, the usability of remote sensing data in land administration in Rwanda is not new. The cadaster was built based on the orthophoto from aerial images acquired using aircraft and satellites in 2008. However, due to the high costs of such projects on a national level, these approaches are difficult to replicate for regular ad-hoc updates. Many changes happen on a daily basis, so up-to-date spatial and non-spatial information is of great importance. The socio-technical assessment completed via this research proves that remotely sensed data has great potential for all land management activities.

Second, the findings of the research show that the current application of remote sensing data and particularly high-resolution UAV images in property valuation for taxation is very limited: traditional methods are still applied. However, as reviewed in ministerial regulation N° 01/MOS/Trans/016, relating to UAVs, the usability of remote aircraft is allowed, and it provides the procedures of how the permit should be issued [44]. Therefore, they can be considered for the tasks for valuation for taxation.

From the interviews, it was confirmed that measuring the property and improvements on the proposed UAV image can reduce fieldwork time significantly. Moreover, from the comparison of the areas delineated over the three imagery data sources, compared to the reference data, the UAV images have the highest spatial resolution. Regarding spatial completeness and up-to-dateness, they have also been proven as very suitable input data. From the interviews, it was revealed that UAVs, as a new data acquisition technology, could be adopted at national and subnational government levels: the national government is the main geospatial data provider in Rwanda. Being low-cost, UAVs can be implemented by local stakeholders to support small-scale mapping with frequent flights. Such actions can contribute to land data gathering in a decentralized way. Similar findings were also found in [45]. Rwanda is one of the most advanced African countries in terms of usage of geospatial data. As such, it seems it would be easier to integrate UAV-based approaches for valuation and land administration. It will be easier to integrate such data into the existing spatial data infrastructure [45].

Although there are many works that outline the benefits of UAV images compared to the other RS techniques, including examples such as - higher spatial and temporal accuracy, and completeness. - the existence of regulations; the need for registered flight companies; license pilots, and so on, present challenges. As was argued in [45], the four main challenges for implementing UAVs as a land tenure data acquisition tool for Rwanda include: (1) Mixed terrain, which means that not all types of UAVs are suitable for flying; (2) Limitations of the current UAV regulations such as flying only in visual sight; (3) Ground truth data collection, especially in an urban environments (existence and reliability of national CORS data); and (4) Software and hardware requirements for data processing. Even cloud-based solutions are being used in many African countries, including Rwanda. However, this approach is often challenging due to the instability of the internet connection. In addition, from the interviews, the lack of human resources and funds for education is another main challenge, especially with regard to data processing and analysis skills.

Many researchers have investigated the optimal photogrammetric workflow configuration to minimize computational costs and reference data collection [46–48]. It is known that obtaining a high-precision GPS and IMU system onboard a UAV can minimize the collection of GCPs, which is usually quite challenging in African countries. Even though UAVs can provide very high-resolution images, the accuracy of captured coordinates may differ from centimeters to several meters, depending on the flight conditions and specific configuration. It has been proven in the research that the number of tie-points has a very significant impact on the correct image orientation process [49]. Based on six experiments in Europe and Africa, the authors can now identify the different flight configurations that will lead to more reliable results for UAV data acquisition. Following the guidelines can assist in achieving the best image outputs that can provide a reliable base for land administration tasks. In addition, based on remotely sensed data, innovative geospatial methods for automatic feature extraction for cadasters have been widely explored by the scientific photogrammetric and computer vision community. For this application, especially for urban areas, UAV data, with its high resolution capability, can outperform other remotely sensed data [50–52].

As mentioned, using remotely sensed data in Rwanda is not new. In terms using it for land valuation for taxation, the existing property valuation law [39] focuses on establishing IRPV and is administratively focused on: methods for valuation, the required data, or the need for standardization. Currently, most of the information is collected from on-ground

field visits. The incompleteness and outdated spatial information from the cadaster in Rwanda have been noticed and reported by other researchers [33,53]. Currently, valuers have to create their own data collection method to execute their duties. There are no guidance documents or rules being followed on how that data can be collected, or which tools could be used, or which remote sensing data can be beneficial to the process. The issue of a lack of data on market transactions results in variations and inconsistencies [33]. In addition, the respondents from RLMUA indicated that the value of land, buildings, and improvements are not separated. Instead, they are recorded as a land value in the LAIS database.

Concerning property tax (fixed asset tax), the taxation system has been developed, but it is still hosted at the national level. According to the law, the property tax should be collected by the districts [29]. Districts have not invested in establishing appropriate revenue collection systems. This results in contracting the Rwanda Revenue Authority as a system holder to collect the tax on behalf of the districts. Currently, self-declaration and ad-hoc systems are being used in Rwanda as the taxation system. In Rwanda, it is the responsibility of the taxpayer to assess their properties and come up with the property value to fill the declaration form. Therefore, in Rwanda, the taxpayers are obliged to register and report their tax obligations to the tax collector [33]. A challenge to the current system is that not all taxpayers comply with this system of self-declaration (self-registration). With the self-assessment system, taxpayers are likely to undervalue their property as the tax is based on the open market value, and the fixed rate is applied. To avoid such results, the proposed remote sensing methods, incorporating UAVs as data acquisition techniques, can assure transparency, higher accuracy, and reliability. Further, to improve precision, more detailed information from terrestrial images or indoor models can also be added. Moreover, it is recommended that in the future this method be combined with additional information such as construction permit data, and indoor information linked to the vertical dimension. This will further improve the accuracy of current valuation.

## 5. Conclusions

RS has been used in Rwanda for the land administration since 2008, and the results of this research show RS continues to grow as an opportunity for land administraiton tasks in that country. In terms of national imagery datasets, there has been no update of the underlying imagery sources since 2008. The current spatial data from RLMUA is outdated and the current methods for spatial data capture are cumbersome and rely on physical inspection using tape measurements, handheld GPS, digital cameras, and so on. Therefore, identifying the best method to update imagery is a pressing concern. Many countries face this challenge.

As such, this research explored the differences between three different remote sensing data acquisition techniques and utimately suggests the usage of the UAV-based approach (not widely available in 2008) for property valuation for taxation purposes, at least for ad-hoc assessments, to be in alignment with the FFPLA principles. That said all RS sources can support assessment of internal factors (parcel size, built-up area; improvement; shape, and type of the subject property) and external factors (accessibility, neighborhood, land use, environment, and utilities). However, the UAV captured RS data was superior in many regards.

In terms of policy, legal, standards, RS approaches, whilst used in land valuation for taxation, could be more systematically applied in the country, especially when it recognised that these land valuation processes, by decree, should be decentralised to the district level (i.e. smaller areas to cover). That said, the lack of finances and trained staff, and only one company being registered to perform UAV flights in the country, present serious inhibitors to the use of UAVs. Many innovative approaches being adopted and applied in practice need initial and substantial governmental support to gain traction, build awareness, and create technical capacity.

Whilst the technical feasibility of the UAV-based approach is clear, workflow and procedures aligned with existing laws and integrated with other land administration processes requires more attention. Moreover, the transferability of the developed approach to other contexts also necessarily requires further investigation. It is suggested that the results of the work here could be the source material for, or at least provide guidance to the ongoing developments of, ISO 19152 LADM, 2nd edition, particularly as that standard will have a dedicated link to property valuation data and processes [54].

This comparison of different remote sensing sources, for land valuation for taxation purposes, can help practitioners, government, and involved institutions to assess the value of remotely sensed data, and the most appropriate sources. Moreover, the work illustrates the benefits of using data with higher spatial and temporal resolution for delivering transparency and a fair land valuation. The use of UAVs can fill the gap between land administration and land management authorities, and eventually strengthen multi-sectoral collaboration.

**Author Contributions:** Conceptualization, M.K., O.G., M.L. and K.A.; writing—original draft, M.K., O.G., M.L. and K.A.; introduction, M.K., R.M.B., O.G., M.L. and K.A.; materials and methods, M.K., O.G., R.M.B., M.L. and K.A.; results, M.K., O.G., R.M.B., M.L. and K.A.; discussion and conclusions, M.K., O.G., R.M.B., M.L., J.Z., K.A. and J.P.; fieldwork, O.G.; review and editing, M.K., R.M.B., O.G., M.L., J.Z., K.A. and J.P. All authors have read and agreed to the published version of the manuscript.

**Funding:** This research received no external funding.

**Institutional Review Board Statement:** The study was conducted according to the guidelines of the Declaration of Helsinki, and approved by the Institutional Review Board.

**Informed Consent Statement:** Informed consent was obtained from all subjects involved in the study.

**Acknowledgments:** The authors acknowledge cadastral experts, including professionals from the Rwanda Land Management Authority, the Rwanda Natural Resources Authority, the Ministry of Infrastructure, district staff, RCMRD, IRPV, RRA, the Organization of Surveyors in Rwanda, and its4land for supporting this study. We acknowledge the land registration department of Rwanda land management authority for providing legal boundaries. We are grateful to ITC for providing image data. We also thank Pix4D for providing us with the research license of Pix4dmapper.

**Conflicts of Interest:** The authors declare no conflict of interest.

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
