# Peer review of "Remote Sensing for Property Valuation: A Data Source Comparison in Support of Fair Land Taxation in Rwanda"

_remotesensing, doi:10.3390/rs13183563_

Round 1

Reviewer 1 Report

The paper analyses the use of remote sensing data (satellite, aerial, UAV) for the purpose of property valuation and taxation in Rwanda. This is an interesting novel application of the remote sensing data and techniques, compared to the traditional methods used in many countries. The paper is well written and well structured. The method for the research is appropriately designed and clearly presented. The results are also clearly presented and support the main hypothesis that workflows relying on remote sensing data, particularly UAV because of its very high resolution, can support property valuation and taxation activities and save time in the process. The paper clearly proved that it is feasible to use remote sensing technologies in these activities. I suggest to add a short discussion how this process can be introduced into the practice through standardized procedures and workflows, what will be the response (acceptance) of the potential users, what are the necessary changes in the legal and institutional framework. The paper analyses the situation in Rwanda, but it would also be interesting to see what is the applicability in other countries. As a future work, the paper mentions a link to indoor information, as this also affects the valuation and taxation of built properties. It would be beneficial to explore this link in the future.

If possible, improve sharpness of figures 5 and 16, the text is somewhat blur. Period at the end of the last sentence is missing (line 718)

Author Response

We provide the attached reply to Reviewer1 and the corrected manuscript.

Reviewer 2 Report

The authors set themselves the task of dealing with a wide range of topics in the paper: “Remote sensing for property valuation: a data source comparison in support of fair land taxation in Rwanda”. Although the focus is on assessing different remote sensing data in support of developing a new approach for property valuation for taxation, basics about the land administration system and the land valuation/taxation in Rwanda is described. General overview of the circumstances in Rwanda provides sufficient background. However, in other parts of the paper it is difficult to find a scientific component.

Conclusion: “The results show that the proposed UAV-based approach in many aspects for valuation and taxation purposes is outperforming compared to satellite and aerial images taken from aircraft.” is generally already well known. Everyone knows that higher resolution images are more preferably. To prove this known fact, no scientific analysis was performed in the paper. Only photos are given for a couple of examples. Although evaluation in terms of accuracy, completeness, up-to-dateness and cost is mentioned, I didn't find it. What was done seems very trivial to me.

The discussion and text around Figure 16 are completely incomprehensible to me. It is not clear to me whether this is a comparison or something else. That part seems unfinished.

The paper has technical shortcomings:

  • the words in the figures are broken
  • the legend in the figures is written in different fonts, mismatched colors ...

The same publication (Fit-for-purpose land administration) was cited several times in the text but was given 3 times with different data in the list of references (1, 32 and 41). The names of the authors are written incorrectly, eg [23] in line 776.

General note: It is not clear to me why "in support of fair land taxation" and not "in support of LAS data improvement". It would be much more cost effective to improve the data for more purposes than just for taxation. I guess that would be FFP approach.

I recommend the authors to omit the "remote sensing" part, and scientifically strengthen the rest.

Author Response

We provide the attached reply to Reviewer2 and the corrected manuscript. 

Round 2

Reviewer 2 Report

Almost all my recommendations were accepted. The paper has been improved. I recommend it for publishing.